# WHEN NEURAL PDE SOLVERS FAIL TO EXTRAPOLATE: SPECTRAL TRUNCATION AND BOUNDARY CONDITION NEGLECT

**Rahul D Ray**
Department of Electronics and Electrical Engineering
Birla Institute of Technology and Science, Pilani – Hyderabad Campus
Hyderabad, India
`f20242213@hyderabad.bits-pilani.ac.in`

## ABSTRACT

Neural solvers for parametric partial differential equations achieve strong empirical accuracy within trained regimes, yet their ability to learn true solution operators remains unclear. We present a controlled diagnostic study of two widely used approaches, the Fourier Neural Operator and Physics Informed Neural Networks, designed to isolate structural extrapolation failures independent of numerical error or optimization instability. For the two dimensional Helmholtz equation, we show that the Fourier Neural Operator generalizes to lower frequency regimes but fails under high frequency extrapolation due to fixed spectral truncation, with prediction errors concentrating in unresolved Fourier modes. For a parametrically scaled Poisson problem, we demonstrate that Physics Informed Neural Networks trained at a single parameter value fail to recover correct amplitude scaling under boundary condition variation. Boundary errors remain small while interior solution errors grow substantially, revealing boundary interior decoupling induced by loss imbalance. Together, these results show that strong in distribution performance can mask reduction of the true operator to a lower capacity surrogate aligned with the training distribution. Our findings highlight the need for diagnostic extrapolation benchmarks and architectures explicitly designed for spectral and parametric generalization.

## 1 INTRODUCTION

Scientific machine learning has emerged as a powerful paradigm for approximating solution operators of parametric partial differential equations (PDEs). Neural operators, particularly the Fourier Neural Operator (FNO) Li et al. (2020); Kovachki et al. (2023), enable learning mappings between infinite-dimensional function spaces and have demonstrated strong empirical performance across fluid dynamics, wave propagation, and frequency-domain problems. In parallel, Physics-Informed Neural Networks (PINNs) incorporate governing equations directly into training and are widely used for boundary value problems. These approaches promise substantial computational acceleration compared to classical solvers while maintaining high accuracy within trained regimes.

Despite rapid progress, a central question remains unresolved: do neural PDE solvers genuinely learn parametric solution operators, or do they interpolate within restricted training distributions shaped by architectural inductive biases? Although numerous works report improved empirical performance through architectural refinements Li et al. (2024); Li & Yang (2025); Ma et al. (2026); You et al. (2025) or robustness-oriented training strategies Liu et al. (2025), systematic diagnostic analyses of structural extrapolation failure remain limited. In particular, the mechanisms underlying high-frequency breakdown in wave equations and parametric boundary-condition scaling failures in physics-informed models are not fully understood.

Fourier Neural Operators are inherently spectral: they parameterize integral kernels in Fourier space and explicitly truncate high-frequency modes Li et al. (2020). This truncation acts as a fixed-bandwidth constraint on the learned operator. Although spectral bias has been discussed in recent

analyses Qin et al. (2024); Fanaskov & Oseledets (2023), most prior work focuses on mitigating performance degradation rather than isolating whether high-frequency extrapolation failure arises from optimization, data scarcity, or intrinsic architectural limitations. Similarly, PINNs enforce PDE residuals and boundary conditions through composite losses, yet imbalance between interior and boundary constraints has been repeatedly observed Maddu et al. (2022); Deguchi & Asai (2023); Gao et al. (2025). Existing remedies emphasize training stabilization, but it remains unclear whether such imbalance induces structural failures under parametric variation.

This work presents a controlled diagnostic study designed to isolate architectural failure modes in neural PDE solvers. Rather than proposing new architectures, we construct analytically tractable experiments that eliminate numerical confounds and optimization instability, allowing direct attribution of failure to inductive bias.

First, we investigate spectral extrapolation in the Fourier Neural Operator for the two-dimensional Helmholtz equation. Training is performed within a restricted wavenumber band, and evaluation spans both lower and higher frequencies. While the FNO generalizes robustly to lower-frequency regimes, it exhibits sharp performance degradation when extrapolating to higher wavenumbers. Fourier-domain error decomposition reveals that prediction errors concentrate in unresolved spectral bands, demonstrating that the learned mapping is effectively a spectrally truncated surrogate rather than the full parametric solution operator.

Second, we examine scale generalization in Physics-Informed Neural Networks using a Poisson boundary value problem with analytically controlled parameter scaling. Training at a single parameter value produces low loss and accurate boundary enforcement. However, under boundary-condition scaling, interior amplitudes fail to adjust appropriately despite small boundary errors. This exposes a structural boundary–interior decoupling induced by loss imbalance: the network minimizes the PDE residual near the training parameter while failing to learn the correct parametric dependence of the solution operator.

Together, these results demonstrate that strong in-distribution performance does not imply operator learning. In both cases, the architectures reduce the true parametric operator to lower-capacity surrogates aligned with the training regime—through fixed spectral bandwidth in FNO and boundary-condition neglect in PINNs. These failures arise under idealized, noise-free, and numerically controlled settings, indicating that they are structural rather than optimization artifacts.

Our findings suggest that scaling model size or increasing training data alone is insufficient to ensure reliable extrapolation. Robust operator learning requires architectural mechanisms that adapt spectral resolution and enforce boundary constraints in a principled manner. Until such mechanisms are incorporated, neural PDE solvers should be interpreted as efficient interpolators within known regimes rather than general-purpose solvers of parametric physical laws.

The primary contributions of this work are:

- A controlled experimental framework that isolates architectural extrapolation failures in neural PDE solvers.
- A spectral-domain diagnostic demonstrating that high-frequency extrapolation failure in FNO arises from fixed spectral truncation.
- A parametric scaling analysis showing that standard PINNs fail to learn correct boundary–interior coupling under boundary-condition variation.
- Mechanistic evidence that interpolation accuracy does not guarantee recovery of the true parametric solution operator.

## 2 RELATED WORK

Neural operators have emerged as a scalable framework for learning solution operators of parametric partial differential equations. The Fourier Neural Operator (FNO) Li et al. (2020); Kovachki et al. (2023) introduced spectral parameterization of integral kernels, enabling efficient approximation of mappings between function spaces. Subsequent work has demonstrated strong empirical performance in wave and frequency-domain applications, including high-frequency seismic wavefield extrapolation Song & Wang (2022), variable-velocity acoustic models Li et al. (2023), elastic wave equations Zhang

et al. (2023), electromagnetic scattering Nikdast & Shishegar (2024), and parametric frequency-domain wave equations with transfer learning Wang et al. (2024). Multi-scale and enhanced variants, such as MscaleFNO You et al. (2024) and feature-integrated FNO architectures Li et al. (2025), attempt to mitigate spectral bias and improve oscillatory regime performance. These studies confirm the practical effectiveness of FNO-based models for Helmholtz-type problems, yet they largely focus on predictive accuracy improvements rather than isolating structural generalization limits.

Recent investigations analyze neural operators from a spectral perspective. Spectral bias in FNO kernels and truncation effects are examined in Qin et al. (2024), while Spectral Neural Operators (SNOs) explore alternative basis representations and aliasing behavior Fanaskov & Oseledets (2023). Physics-informed neural operators (PINO) integrate governing equations to enhance resolution robustness Li et al. (2024), and multiphysics or physics-knowledge-augmented frameworks aim to improve data efficiency and out-of-distribution (OOD) generalization Ma et al. (2026). Additional efforts address OOD robustness through dual-branch architectures Li & Yang (2025), risk-averse stochastic optimization Liu et al. (2025), and self-supervised operator learning You et al. (2025). While these works propose architectural refinements or training objectives to enhance empirical robustness, they do not explicitly disentangle whether extrapolation failure originates from optimization dynamics, insufficient data coverage, or intrinsic architectural bandwidth constraints. Our work addresses this gap by constructing controlled frequency-scaling experiments that directly attribute high-frequency breakdown to fixed spectral truncation, independent of training instability.

Learning solution operators across parameter spaces has also been studied via physics-informed DeepONets Wang et al. (2021) and unified perspectives bridging PINNs and neural operators Dai et al. (2026); Zhang et al. (2025). These works emphasize operator-level generalization and improved parametric performance, often reporting strong interpolation results across sampled regimes. However, evaluation typically remains confined to distributions represented during training. Our analysis demonstrates that strong in-distribution accuracy does not imply recovery of the true parametric operator, particularly when extrapolation requires spectral resolution or amplitude scaling beyond the training bandwidth.

Physics-Informed Neural Networks (PINNs) enforce governing equations and boundary conditions through composite loss functions. Numerous studies document imbalance between PDE residual and boundary losses and propose weighting strategies to address this pathology, including inverse Dirichlet weighting Maddu et al. (2022), dynamic gradient normalization Deguchi & Asai (2023), adaptive loss balancing Gao et al. (2025), boundary-aware feature enforcement Jahani-Nasab & Bijarchi (2024), and stabilized multiscale formulations for advection-dominated problems Hsieh & Huang (2024). These approaches improve optimization stability and boundary satisfaction, yet evaluation is typically performed at fixed parameter settings. Our work differs by examining parametric boundary-condition scaling under analytically controlled conditions and showing that even when boundary errors remain small, interior amplitudes fail to scale correctly. This reveals a structural boundary–interior decoupling that persists despite low training loss and is not captured by conventional diagnostics.

Overall, prior literature prioritizes architectural innovation and empirical performance gains. What remains insufficiently understood is whether current neural PDE solvers genuinely learn parametric solution operators or instead approximate lower-capacity surrogates aligned with the training distribution. By isolating spectral truncation in FNO and boundary-condition neglect in PINNs under controlled extrapolation settings, our work provides mechanistic evidence that interpolation success does not guarantee operator learning. This diagnostic perspective complements existing advancements and identifies fundamental architectural limitations that must be addressed for reliable extrapolation beyond trained regimes.

## 3 SPECTRAL EXTRAPOLATION FAILURE IN FOURIER NEURAL OPERATORS

### 3.1 PROBLEM SETUP: HELMHOLTZ EQUATION

We consider the two-dimensional Helmholtz equation

$$(\Delta + k^2)u = f \quad \text{in } \Omega = [0, 1]^2, \tag{1}$$

with homogeneous Dirichlet boundary conditions. The domain is discretized on a uniform $64 \times 64$ grid using second-order finite differences, yielding the sparse linear system

$$(L + k^2 I)\mathbf{u} = \mathbf{f}, \tag{2}$$

where $L$ denotes the discrete Laplacian constructed via Kronecker products. The forcing $f$ is generated as a sum of three Gaussian blobs with randomized centers, amplitudes, and widths. Solutions are computed exactly using a direct sparse solver (see Appendix A.1 and Appendix 3.2).

The training set consists of 500 samples with wavenumbers uniformly drawn from $k \in [15\pi, 20\pi]$. Generalization is evaluated at $k \in \{10\pi, 12.5\pi, 17.5\pi, 22.5\pi, 25\pi\}$, spanning both lower- and higher-frequency regimes relative to training.

## 3.2 Forcing Function Generation for the Helmholtz Experiments

Each forcing field $f(x, y)$ is generated as a sum of three spatially localized Gaussian components of the form

$$f(x, y) = \sum_{i=1}^{3} a_i \exp\left(-\frac{(x - x_i)^2 + (y - y_i)^2}{2\sigma_i^2}\right). \tag{3}$$

The amplitudes $a_i$, centers $(x_i, y_i)$, and widths $\sigma_i$ are independently sampled from uniform distributions over fixed ranges chosen to ensure sufficient diversity without introducing numerical stiffness. This construction yields forcing terms that are non periodic, spatially heterogeneous, and possess broad spectral support.

For each forcing realization, solutions are computed for a single value of the wavenumber $k$. Training data consist of 500 such pairs with $k$ uniformly sampled from the interval $[15\pi, 20\pi]$. Test data are generated independently for each evaluation wavenumber to avoid leakage of spatial patterns from the training set.

## 3.3 Fourier Neural Operator Architecture

The Fourier Neural Operator used in this study follows the standard two dimensional formulation. The input consists of two channels corresponding to the forcing field and the scalar wavenumber, broadcast spatially. These inputs are first lifted into a higher dimensional feature space using a pointwise linear transformation.

Each spectral convolution layer performs the following operations. Given an input feature map $v(x)$, a two dimensional discrete Fourier transform is applied. Learned complex valued weights are then applied to a fixed set of low frequency Fourier modes satisfying $|\xi_1|, |\xi_2| \leq m$ with $m = 16$. All higher frequency modes are passed through unchanged. The modified spectrum is then transformed back to physical space using an inverse Fourier transform, and combined with a pointwise linear residual connection.

Four such spectral convolution layers are stacked, followed by a projection back to the scalar output field. The total parameter count of the model is approximately 4.2 million. This architecture imposes an explicit spectral bandwidth limitation that plays a central role in the observed extrapolation behavior.

## 3.4 Extrapolation Performance Across Wavenumbers

The baseline CNN fails catastrophically at all test wavenumbers, with relative errors exceeding $100\%$, confirming that purely local operators are insufficient for oscillatory elliptic PDEs. In contrast, the FNO exhibits strongly asymmetric extrapolation behavior. For wavenumbers below the training range $(10\pi, 12.5\pi)$, it achieves low mean relative error ($\approx 1.1$) with small variance, indicating robust extrapolation to spectrally simpler regimes. Performance at the in-distribution test point $(17.5\pi)$ degrades moderately.

High-frequency extrapolation $(22.5\pi, 25\pi)$ produces a sharp increase in both mean error and variance, with standard deviations comparable to or exceeding the mean (Fig. 2b). This instability indicates that the FNO operates at the boundary of its learned representation when required to resolve higher-frequency solutions.

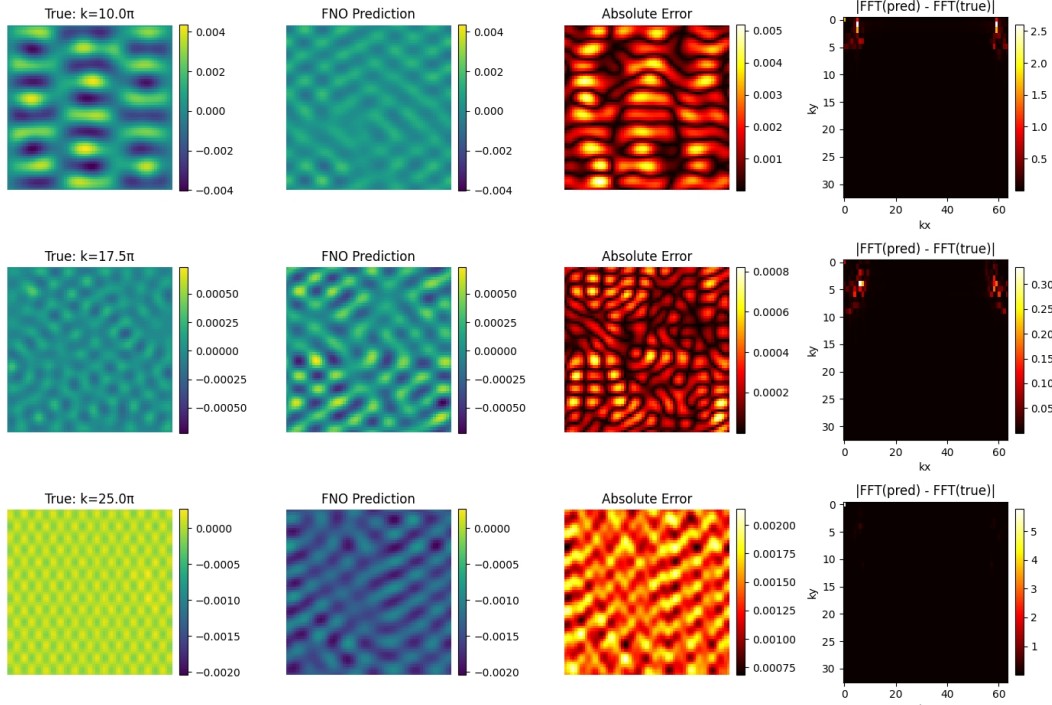

Figure 1: Appendix A. Sample solutions and Fourier-domain errors for the Fourier Neural Operator applied to the two-dimensional Helmholtz equation. The figure contains 12 panels arranged internally, with rows corresponding to test wavenumbers $k = 10\pi$, $17.5\pi$, and $25\pi$, and columns showing the true solution, FNO prediction, absolute error in physical space, and Fourier-domain error magnitude $|\mathcal{F}(\hat{u}) - \mathcal{F}(u)|$. As the wavenumber increases beyond the training range, prediction errors concentrate at finer spatial scales, indicating missing high-frequency spectral content.

## 3.5 SPECTRAL ERROR DECOMPOSITION

To localize the source of extrapolation failure, we analyze predictions in the Fourier domain. For each test case, we compute Fourier transforms of predicted and exact solutions and evaluate the mean absolute error binned by radial frequency magnitude $|\xi|$:

$$\epsilon(|\xi|) = \mathbb{E}\left[|\mathcal{F}(u_{\mathrm{pred}})(\xi) - \mathcal{F}(u_{\mathrm{true}})(\xi)|\right]. \qquad (4)$$

For wavenumbers within or below the training band, spectral errors remain uniformly low across frequencies. In contrast, high-frequency extrapolation leads to errors that grow monotonically with $|\xi|$, concentrating in mid-to-high spectral bands beyond the retained Fourier modes (Fig. 2c). This behavior demonstrates that the failure is structural rather than uniform: the FNO is unable to synthesize the higher-frequency spectral content required by solutions at larger $k$ (see Appendix 3.5).

To identify the origin of extrapolation errors, predictions are analyzed in Fourier space. For each test sample, the discrete Fourier transforms of the predicted and exact solutions are computed. The absolute difference between spectra is then binned according to radial frequency magnitude. The reported spectral error metric is

$$\epsilon(|\xi|) = \mathbb{E}\left[|\mathcal{F}(u_{\mathrm{pred}})(\xi) - \mathcal{F}(u_{\mathrm{true}})(\xi)|\right], \qquad (5)$$

where the expectation is taken over samples at fixed wavenumber. This analysis reveals whether errors are uniformly distributed or concentrated in specific frequency bands. Qualitative spatial-domain examples and corresponding Fourier-domain diagnostics supporting this analysis are shown in Fig. 1 and Fig. 3.

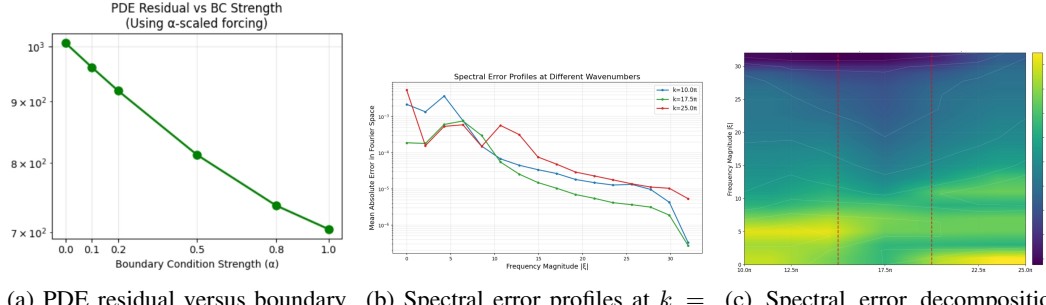

(a) PDE residual versus boundary condition strength.

(b) Spectral error profiles at $k = 10\pi$, $17.5\pi$, and $25\pi$.

(c) Spectral error decomposition across test wavenumbers.

Figure 2: Diagnostic evidence of failure mechanisms in neural PDE solvers.

# 4 SCALE GENERALIZATION FAILURE IN PHYSICS-INFORMED NEURAL NETWORKS

## 4.1 PARAMETRIC POISSON PROBLEM FORMULATION

We consider the Poisson equation

$$\nabla^2 u = f \quad \text{in } \Omega = [0,1]^2, \tag{6}$$

with Dirichlet boundary conditions $u|_{\partial\Omega} = g$. To study parametric generalization, we introduce a scalar parameter $\alpha$ that linearly scales both forcing and boundary data:

$$\nabla^2 u_\alpha = \alpha f_{\text{base}}, \quad u_\alpha|_{\partial\Omega} = \alpha g_{\text{base}}. \tag{7}$$

The base solution $u_{\text{base}}(x,y) = \sin(\pi x)\sin(\pi y)$ yields $f_{\text{base}} = 2\pi^2 \sin(\pi x)\sin(\pi y)$, with homogeneous Dirichlet boundary conditions. This construction enforces exact PDE consistency and admits the closed-form solution $u_\alpha = \alpha u_{\text{base}}$, providing a controlled setting in which deviations from linear scaling directly indicate generalization failure (Appendix A.2).

## 4.2 PINN ARCHITECTURE AND LOSS CONSTRUCTION

The solution $u(x,y;\theta)$ is approximated using a fully connected multilayer perceptron with $\tanh$ activations. Training minimizes the composite loss

$$\mathcal{L}(\theta) = \lambda_{\text{BC}}\mathcal{L}_{\text{BC}} + \lambda_{\text{PDE}}\mathcal{L}_{\text{PDE}}, \tag{8}$$

with

$$\mathcal{L}_{\text{BC}} = \frac{1}{N_{\text{BC}}} \sum_i \left| u(\mathbf{x}_i^{\text{BC}};\theta) - u_\alpha(\mathbf{x}_i^{\text{BC}}) \right|^2, \tag{9}$$

$$\mathcal{L}_{\text{PDE}} = \frac{1}{N_{\text{int}}} \sum_j \left| -\nabla^2 u(\mathbf{x}_j^{\text{int}};\theta) - f_\alpha(\mathbf{x}_j^{\text{int}}) \right|^2. \tag{10}$$

Training is performed exclusively at $\alpha = 1.0$ with equal loss weights. Boundary and interior collocation points are sampled on a coarse $10 \times 10$ grid, and optimization is carried out for 10,000 epochs.

## 4.3 BASELINE CONVOLUTIONAL MODEL

As a control, we employ a minimal convolutional neural network consisting of a single $3 \times 3$ convolutional layer with no nonlinearity. This model contains 18 trainable parameters and operates purely locally. Its purpose is not to provide a competitive baseline, but to verify that any observed extrapolation capability in the FNO is nontrivial and not shared by generic local operators.

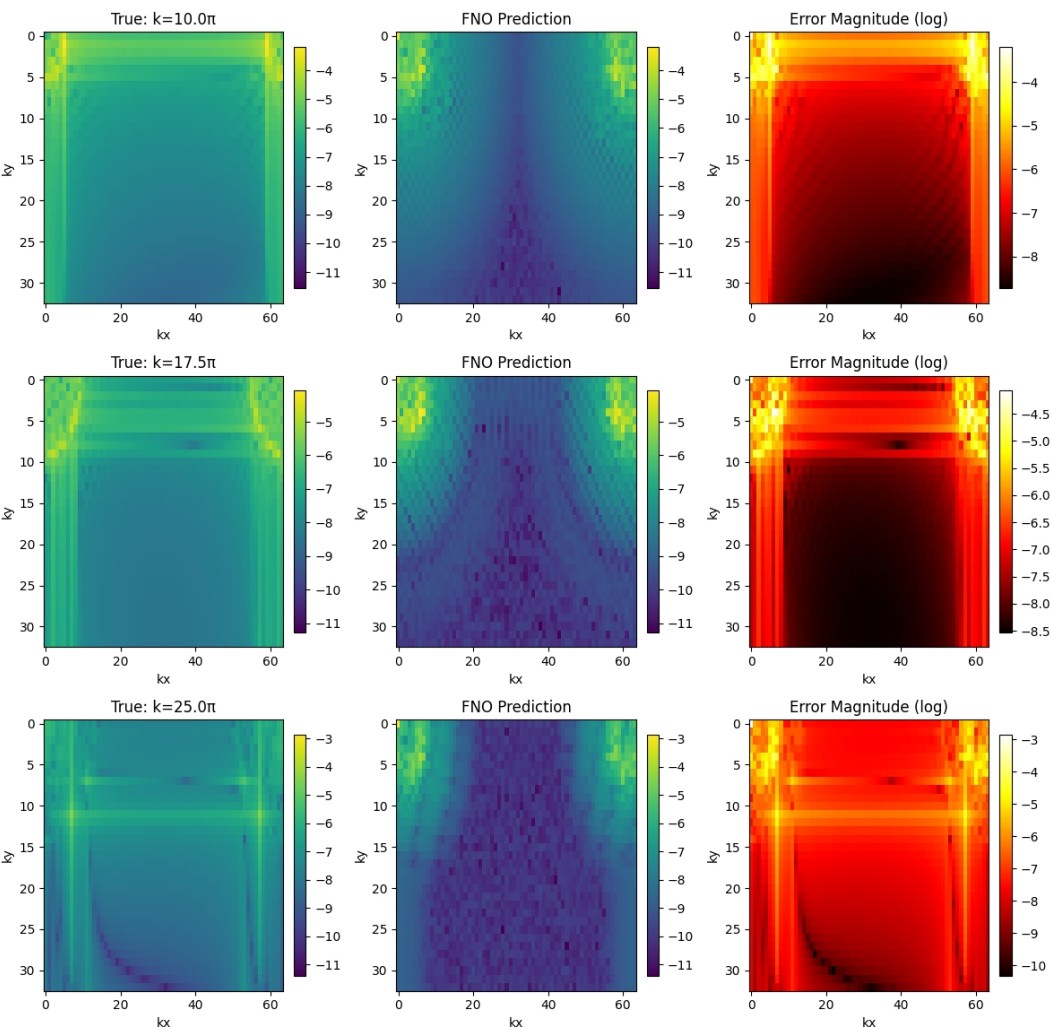

Figure 3: Fourier-domain comparison on a logarithmic scale for increasing test wavenumbers. The figure contains 9 panels arranged internally, with rows corresponding to $k = 10\pi$, $17.5\pi$, and $25\pi$, and columns showing the true Fourier spectrum, the Fourier Neural Operator prediction, and the error magnitude in the frequency domain. While low-frequency components are accurately captured across all cases, high-frequency discrepancies grow substantially for $k = 25\pi$, confirming that extrapolation failure is dominated by missing high-frequency spectral modes.

## 4.4 Evaluation Metrics for Helmholtz Experiments

Model performance is quantified using the relative $L^2$ error

$$\epsilon_{\text{rel}} = \frac{\|u_{\text{pred}} - u_{\text{true}}\|_2}{\|u_{\text{true}}\|_2}. \tag{11}$$

For each wavenumber, mean and standard deviation are computed over multiple independent test samples. High variance is interpreted as instability under extrapolation rather than stochastic noise.

## 4.5 Generalization Under Boundary Condition Scaling

Generalization is evaluated at $\alpha \in \{0.0, 0.1, 0.2, 0.5, 0.8, 1.0\}$ using a disjoint test grid. Although the model achieves a low training loss at $\alpha = 1.0$, it fails catastrophically at other values. The relative $L^2$ error increases monotonically as $\alpha$ deviates from the training condition, exceeding 50 at $\alpha = 0.0$.

While boundary condition mean squared error remains small across all $\alpha$, interior errors and PDE residuals remain large and decrease only as $\alpha$ approaches the training value. This indicates memorization of a single solution instance rather than learning the parametric solution operator (Appendix A.3).

### 4.6 MECHANISM OF BOUNDARY CONDITION NEGLECT

This failure arises from an imbalance in how the loss terms constrain the solution. The boundary loss acts on a lower-dimensional manifold, whereas the PDE residual is enforced throughout the interior domain and dominates optimization. Consequently, the network converges to a shortcut solution of the form

$$u_\theta \approx c\, u_{\text{base}}, \tag{12}$$

with a fixed amplitude $c$ that approximately satisfies the PDE residual near $\alpha = 1$ but does not scale with $\alpha$ (Appendix A.4).

As $\alpha$ varies, the interior solution fails to adjust globally, leading to incorrect amplitude scaling and large relative errors despite small boundary errors. Overall, these results demonstrate a fundamental limitation of standard PINN formulations for parametric elliptic problems: minimizing PDE residuals and boundary errors at a single parameter value does not ensure learning the correct solution operator across the parameter space (Appendix A).

## 5 DISCUSSION: ARCHITECTURAL LIMITS OF NEURAL PDE SOLVERS

This work provides controlled diagnostic evidence that widely used neural PDE solvers can fail for structural reasons that are not attributable to data scarcity, numerical error, or optimization instability. In the Fourier Neural Operator (FNO), extrapolation is fundamentally constrained by fixed spectral bandwidth. Because only a finite set of low-frequency modes is parameterized, the learned mapping effectively becomes a spectrally truncated approximation of the true solution operator. While this surrogate performs reliably within and below the training frequency regime, it breaks down when higher-frequency content is required. Fourier-domain error decomposition shows that failure is not uniform but concentrated in unresolved spectral bands, indicating an intrinsic representational limitation rather than stochastic degradation.

In Physics-Informed Neural Networks (PINNs), failure arises through a distinct but conceptually related mechanism. Soft enforcement of boundary conditions within a composite loss induces an imbalance between lower-dimensional boundary constraints and interior PDE residual minimization. Under single-parameter training, the optimizer converges to a shortcut solution that approximately satisfies the differential operator near the training parameter while failing to learn the correct parametric dependence of the solution. Boundary errors remain small under parameter scaling, yet interior amplitudes fail to adjust appropriately. This boundary–interior decoupling demonstrates that minimizing PDE residuals and boundary losses at a single parameter value does not imply recovery of the true parametric solution operator.

Despite architectural differences, both failures share a common cause: strong in-distribution performance masks reduction of the full operator to a lower-capacity surrogate aligned with the training distribution. These pathologies remain invisible under standard evaluation protocols and only emerge through targeted extrapolation diagnostics. The results therefore suggest that scaling model size, training data, or optimization effort alone is insufficient to ensure operator-level generalization. Robust neural PDE solvers will likely require adaptive spectral representations, parameter-aware conditioning mechanisms, and principled or hard enforcement of boundary constraints.

Several limitations define the scope of this study. The experiments are restricted to elliptic equations on simple geometries with homogeneous Dirichlet boundary conditions and moderate spatial resolution. Models are intentionally trained under minimal and controlled protocols—fixed spectral truncation for FNO and single-parameter training for PINNs to isolate structural behavior rather than explore mitigation strategies such as adaptive loss weighting, curriculum learning, multiparameter conditioning, or dynamic spectral refinement. The analysis also assumes deterministic, noise-free data. While quantitative behavior may differ in more complex geometries, higher dimensions, or time-dependent systems, the controlled failures demonstrated here indicate that architectural inductive bias alone can induce systematic extrapolation breakdown. Addressing these limitations will

require explicitly generalization-aware operator design rather than purely empirical performance optimization.

## 6 CONCLUSION

This work presents a focused diagnostic study of generalization failures in neural PDE solvers. Through controlled experiments on the Helmholtz and Poisson equations, we demonstrate that Fourier Neural Operators suffer from spectral extrapolation breakdown, while Physics-Informed Neural Networks exhibit scale generalization failure due to boundary condition neglect. Together, these results show that successful interpolation does not imply operator learning. Our analysis highlights the necessity of diagnostic benchmarks that probe extrapolation, rather than relying solely on in-distribution accuracy. Addressing these failure modes will require architectural innovations that couple spectral adaptivity with principled constraint enforcement. Until such mechanisms are incorporated, neural PDE solvers should be viewed as efficient interpolators within known regimes rather than general-purpose solvers of parametric physical laws.

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

## A  DETAILED NUMERICAL AND EXPERIMENTAL METHODOLOGY

This appendix details the numerical solvers, data generation procedures, model architectures, training protocols, and diagnostic metrics used throughout the study. Its purpose is to ensure transparency and reproducibility, and to establish that the observed failure modes stem from intrinsic architectural limitations rather than numerical artifacts, inconsistent problem formulations, or optimization instability.

### A.1  NUMERICAL DISCRETIZATION OF THE HELMHOLTZ EQUATION

We consider the two dimensional Helmholtz equation

$$(\Delta + k^2)u = f \tag{13}$$

on the unit square domain $\Omega = [0, 1]^2$ with homogeneous Dirichlet boundary conditions. The domain is discretized using a uniform Cartesian grid with $N = 64$ points in each spatial direction. Let

$h = 1/(N+1)$ denote the grid spacing. The Laplacian operator is approximated using second order central finite differences.

In one dimension, the discrete second difference operator $D_2 \in \mathbb{R}^{N \times N}$ is given by

$$D_2 = \frac{1}{h^2} \begin{bmatrix} -2 & 1 & 0 & \cdots & 0 \\ 1 & -2 & 1 & \cdots & 0 \\ 0 & 1 & -2 & \cdots & 0 \\ \vdots & \vdots & \vdots & \ddots & 1 \\ 0 & 0 & 0 & 1 & -2 \end{bmatrix}, \tag{14}$$

where boundary values are implicitly set to zero to enforce Dirichlet conditions. The two dimensional Laplacian is constructed via the Kronecker sum

$$L = D_2 \otimes I + I \otimes D_2, \tag{15}$$

where $I$ denotes the $N \times N$ identity matrix and $\otimes$ denotes the Kronecker product.

For each forcing function $f$ and wavenumber $k$, the discrete system

$$(L + k^2 I)\mathbf{u} = \mathbf{f} \tag{16}$$

is solved exactly using a sparse direct solver. This ensures that numerical error from the PDE solver is negligible compared to the approximation error introduced by the neural models. All reported results therefore reflect model behavior rather than discretization artifacts.

## A.2 PARAMETRIC POISSON PROBLEM FORMULATION

For the PINN experiments, we consider the Poisson equation

$$-\nabla^2 u = f \tag{17}$$

on $\Omega = [0,1]^2$ with Dirichlet boundary conditions. To study scale generalization, we introduce a scalar parameter $\alpha$ that scales both the forcing term and the boundary condition. The base solution is defined as

$$u_{\text{base}}(x, y) = \sin(\pi x)\sin(\pi y), \tag{18}$$

which satisfies homogeneous Dirichlet boundary conditions. The corresponding forcing term is

$$f_{\text{base}} = 2\pi^2 \sin(\pi x)\sin(\pi y). \tag{19}$$

For each $\alpha$, the exact solution is $u_\alpha = \alpha u_{\text{base}}$, and the forcing term is scaled accordingly. This construction ensures exact PDE consistency and eliminates ambiguity regarding the correct parametric dependence.

## A.3 EVALUATION OF PARAMETRIC GENERALIZATION

Generalization is evaluated on a set of $\alpha$ values disjoint from the training condition. For each $\alpha$, we compute the relative $L^2$ error over the domain, the boundary mean squared error, and the PDE residual mean squared error. Reporting these metrics separately allows boundary satisfaction to be analyzed independently from interior solution consistency, which is essential for diagnosing failure modes in physics-informed formulations.

Although the model achieves low training loss at $\alpha = 1.0$, it fails to generalize to unseen parameter values. Boundary errors remain uniformly small across all tested $\alpha$, indicating that the network continues to match the prescribed Dirichlet data even under parameter variation. In contrast, interior solution errors and PDE residuals grow rapidly as $\alpha$ deviates from the training value, with the largest discrepancies observed at extreme extrapolation points.

This divergence reveals that the network has not learned the correct parametric dependence of the solution operator. Instead, it converges to a solution that satisfies boundary constraints while approximately minimizing the PDE residual only near the training parameter. As a result, the learned representation fails to scale globally with $\alpha$, leading to severe interior inaccuracies despite superficially correct boundary behavior. This decoupling between boundary enforcement and interior consistency highlights a fundamental limitation of the PINN formulation under single-parameter training.

### A.4 Interpretation of Boundary Condition Neglect

The observed behavior arises from an imbalance between boundary and interior constraints during optimization. The boundary loss acts only on a lower-dimensional subset of the domain, whereas the PDE residual is enforced throughout the interior and therefore dominates the gradient signal during training. As a consequence, the optimizer prioritizes reducing the interior residual near the training parameter, even if doing so requires distorting the global scaling of the solution.

This imbalance encourages the network to converge to shortcut solutions that approximately satisfy the differential operator while decoupling the interior solution from the boundary data under parameter variation. In such solutions, the learned field retains a fixed amplitude determined by the training condition and fails to respond appropriately as the boundary forcing is scaled. The resulting violation of the causal relationship between boundary conditions and interior values persists despite low overall training loss and visually plausible solutions. Without explicit parametric evaluation, this failure mode would remain hidden, underscoring the inadequacy of standard loss formulations for diagnosing true operator generalization.

## B Statement on the Use of Large Language Models

Large Language Models were used in a limited capacity to refine grammar and improve sentence clarity after the initial draft was completed. All research ideas, methodologies, experiments, analyses, and conclusions were developed entirely by the authors. The LLM did not contribute to scientific content, experimental decisions, or academic judgment. This use does not compromise the originality or academic integrity of the work.

