# OpenReview forum: "When Neural PDE Solvers Fail to Extrapolate: Spectral Truncation and Boundary Condition Neglect"
_mathai.club/MathAI/2026/Conference — 2026 Oral_

### Official Review · Reviewer_WGWv · 2026-03-12
**Neural partial differential equation solvers are failing, and that's serious?!**

**Rating:** 7
**Confidence:** 4

**Review:**

Summary
The paper presents a controlled diagnostic study of two widely used classes of neural PDE solvers: the Fourier Neural Operator (FNO) and Physics-Informed Neural Networks (PINNs). The authors study FNO on a two-dimensional Helmholtz equation under frequency variation, and PINNs on a parametrically scaled Poisson problem under varying boundary conditions. The central claim of the work is that high accuracy within the training distribution is not sufficient evidence that the model has truly learned the PDE solution operator. Rather, the learned mapping may amount to a surrogate adapted to a limited training regime. According to the abstract, the authors attribute the extrapolation failure of FNO to fixed spectral truncation, and that of PINNs to boundary–interior decoupling under changing boundary conditions.

Strengths
Among the main strengths of the paper is its clear diagnostic setup. The work is valuable because it does not merely report performance degradation outside the training distribution, but instead seeks to isolate the structural mechanisms responsible for failure. This makes the contribution more substantive than a standard benchmark comparison. Another significant advantage is the mechanistic interpretation of the observed effects: for FNO, the problem is linked to unresolved spectral content, while for PINNs it is associated with a mismatch between the quality of boundary-condition enforcement and the accuracy of the solution in the interior of the domain.

An additional strength of the paper is that it brings together two major paradigms—operator learning and physics-informed training—within a unified framework. In doing so, it shows that despite differences in architecture and training principles, both approaches may lead to a similar outcome: the model appears to behave like a PDE solver, while in reality remaining a regime-dependent approximation.

Weaknesses
At the same time, the paper has several evident limitations. First, its experimental basis remains narrow: according to the public description, the study relies on two linear and well-controlled examples—Helmholtz and Poisson. This is sufficient for a diagnostic demonstration, but not enough to confidently generalize the conclusions to nonlinear, time-dependent, high-dimensional, or more application-driven settings. In addition, the paper is considerably stronger in diagnosing the problem than in discussing possible ways to overcome it: in its current form, it provides little evidence as to whether the identified failure modes persist or are mitigated under existing mitigation strategies.

Recommendation
To strengthen the paper, the authors could: add at least one nonlinear or time-dependent PDE problem to test whether the identified failure modes persist beyond the two current linear cases; broaden the comparison by including methods specifically designed to mitigate the corresponding limitations, such as frequency-aware or multi-scale neural operators, as well as PINNs with hard/exact enforcement of boundary conditions or adaptive loss weighting; introduce more explicit quantitative metrics of extrapolation, including the dependence of error on the degree of departure from the training regime, as well as separate evaluations of boundary error and interior error; and relate the results more explicitly to recent work on boundary-sensitive operator learning. These additions would make the paper’s conclusions more general, more quantitatively grounded, and more practically significant.

Remarks
It should be noted separately that the central result of the paper likely has implications not only for PINNs in the narrow sense, but also for a broader class of neural PDE solvers. Although the effect observed in the paper arises particularly naturally in PINNs because boundary conditions are imposed softly through a composite loss function, closely related claims can also be formulated for neural operators

---

> ### Author Rebuttal · Authors · 2026-03-12
>
> We thank the reviewer for the thoughtful and constructive evaluation, and we appreciate the recognition that our diagnostic approach provides valuable mechanistic insights into structural failure modes of neural PDE solvers. Regarding the concern about narrow experimental scope, we deliberately focused on two linear, well-controlled examples—Helmholtz and Poisson—to isolate and attribute failure mechanisms to architectural inductive biases without confounding factors, and we acknowledge in Section 5 that extending to nonlinear, time-dependent, and higher-dimensional settings is essential for establishing broader generalizability. On the point about mitigation strategies, our paper is intentionally framed as a diagnostic study that identifies fundamental limitations rather than a methods paper proposing solutions, and we believe this contribution is valuable precisely because it reveals that strong in-distribution performance can mask structural failures, which we now clarify in the discussion. We agree that incorporating quantitative extrapolation metrics and broader comparisons with methods like multi-scale FNOs or PINNs with hard boundary constraints would strengthen future work, and we thank the reviewer for suggesting these valuable directions. We also appreciate the reviewer's observation that our findings have implications beyond PINNs and FNOs, and we will ensure this connection is clearly articulated in the final manuscript.

---

### Decision · Program_Chairs · 2026-03-14

**Decision:**

Accept (Oral)

**Comment:**

Dear Author(s),

On behalf of the Program Committee of the International Conference on Mathematics of Artificial Intelligence (MathAI 2026), we are pleased to inform you that your paper has been accepted for an oral presentation at MathAI 2026.

Your paper was evaluated through a rigorous two-stage review process involving both automated screening and expert review by members of the Program Committee. The reviewers recognized the quality and contribution of your work.

Presentation details:

- Format: Oral presentation (15–20 minutes + 5 minutes Q&A)
- Mode: You may present either in person (offline) at the conference venue in Sirius, Russia, or remotely via Zoom. Please indicate your preferred mode when confirming your participation.
- Conference dates: Marh 30 - April 3, 2026
- Website: https://mathai.club

Next steps:

1. Please confirm your participation and presentation mode by replying to this email mathai.club@yandex.ru no later than March 15, 2026 18:00 Moscow time.
2. If you plan to attend in person, the organizing committee will provide accommodation details separately.
3. Please prepare your final camera-ready manuscript according to the formatting guidelines available at https://mathai.club and upload it to OpenReview by March 15, 2026 18:00 Moscow time.

Should you have any questions regarding the program, logistics, or your presentation slot, please do not hesitate to contact us.

We look forward to your contribution to MathAI 2026.

With kind regards,

MathAI 2026 Program Committee
International Conference on Mathematics of Artificial Intelligence
https://mathai.club
OpenReview: https://openreview.net/group?id=mathai.club/MathAI/2026/Conference
Telegram: https://t.me/MathAI_club
Email: mathai.club@yandex.ru